# Lightweight Scene Text Recognition Based on Transformer

**DOI:** 10.3390/s23094490

**Published:** 2023-05-05

**Authors:** Xin Luan, Jinwei Zhang, Miaomiao Xu, Wushouer Silamu, Yanbing Li

**Affiliations:** 1College of Information Science and Engineering, Xinjiang University, No. 777 Huarui Street, Urumqi 830017, China; 2Xinjiang Laboratory of Multi-Language Information Technology, Xinjiang University, No. 777 Huarui Street, Urumqi 830017, China; 3Xinjiang Multilingual Information Technology Research Center, Xinjiang University, No. 777 Huarui Street, Urumqi 830017, China

**Keywords:** scene text recognition, transformer, attention mechanism

## Abstract

Scene text recognition (STR) has been a hot research field in computer vision, aiming to recognize text in natural scenes using computers. Currently, attention-based encoder–decoder frameworks struggle to precisely align feature regions with the target object when dealing with complex and low-quality images, a phenomenon known as attention drift. Additionally, with the rise of Transformer, the increasing size of parameters results in higher computational costs. In order to solve the above problems, based on the latest research results of Vision Transformer (ViT), we utilize an additional position-enhancement branch to alleviate attention drift and dynamically fused position information with visual information to achieve better recognition accuracy. The experimental results demonstrate that our model achieves a 3% higher average recognition accuracy on the test set compared to the baseline. Meanwhile, our model maintains the advantage of a small number of parameters and fast inference speed, achieving a good balance between accuracy, speed, and computational load.

## 1. Introduction

Optical character recognition (OCR) [1,2] refers to the printed characters from the paper document recognition proposed by German scientist Tausheck in 1929. With the development of OCR technology, people are satisfied with the recognition of the text in the document or the book and focus on text recognition in the actual scene, which is called scene text recognition (STR).

STR is the expansion and extension of traditional OCR technology in natural scene images and has broad application prospects. STR, such as automatic driving and license plate recognition, has been widely used daily [3,4,5,6]. Natural scene text extraction technology can be divided into two sub-tasks: scene text detection [7,8] and scene text recognition [9,10]. The task of scene text detection is to locate relevant characters in the natural scene. Scene text recognition is converting the text in the cropped picture into the corresponding characters. Although STR began to be active at the end of the 1990s, with the rise of deep neural networks in recent years, its accuracy has reached the commercial level.

Inspired by the machine translation of natural language processing [11], the coder–decoder framework of attention mechanism is widely used in text recognition in natural scenes and has significantly improved the performance [12]. This technology learns the mapping relationship between the input image and output sequence in a data-driven way. The model’s encoder usually uses CNN/Transformer to convert the image into a sequence of feature vectors. Each feature vector represents an area of the image called the attention area. The decoding part’s attention network (AN) [13] first calculates the alignment factor by referring to the history of the target character and the encoded feature vector to generate the composite vector to achieve the alignment between the attention area and the corresponding label. Then, the target character is generated according to the synthetic vector and the history of the target character. However, in dealing with some complex and low-quality images, such as image distortion and blurring, the relevant research [14] found that the existing attention-mechanism methods do not perform well in this situation. One of the main reasons is that the existing methods can not obtain an accurate alignment between the feature area and the target for such an image, which is called “attention drift”.

In order to improve the above problems, some studies [15,16] have used character position and semantic information to improve the model’s accuracy. However, with the development of deep learning, the number of layers of Resnet and Transformer is becoming deeper and deeper, more and more information is added, and the cost of the model has doubled. Most STR models ignore the speed and operation cost, which leads to a competitive performance of the model but a high consumption cost. Therefore, the speed and operation cost of the model are equally important [17]. An ideal model should not focus solely on the model’s performance, regardless of the cost required by the model.

In this work, we balanced recognition accuracy, model speed, and cost. Based on Vit-tiny, we built a **l**ightweight **s**cene **t**ext **r**ecognition model (LSTR) and designed position-enhancement and visual-enhancement modules, allowing the model to maintain a fast inference speed while achieving higher accuracy at a lower cost. The contributions are as follows:

1. The LSTR model improves the problem of attention drift and uses position information to align target labels with visual features, which improves the model performance.

2. We have achieved a balance in the size and accuracy of the model with good practical application ability and model parameters of only 7 m that can thoroughly learn the image features, obtain a good recognition effect, and perform at, on average, 13.6 ms per text image.

3. The LSTR is compared with the latest model, and the results show that the performance of our model is better than the existing methods.

## 2. Related Work

Traditional natural scene text recognition methods [18,19,20] focus on ordinary text, mostly adopting the bottom-up method, detecting a single character using a sliding window, then integrating these characters into the output text [21]. These methods rely on accurate character classifiers and cannot deal with complex situations well.

The modern method [22,23] is top-down and regards scene text recognition as a sequence-prediction problem. This method conducts end-to-end prediction for text sequences without single-character detection. Most of the recent work has focused on irregular text recognition because the text in natural scenes has a complex background, such as fuzziness, low contrast, and a variety of font styles. Therefore, the text-recognition algorithm must consider the image’s complex background and text diversity. We divide the text-recognition models of modern scenes into three categories:

Segmentation-based: The segmentation-based approach [24] treats STR as a semantic segmentation problem. Ca-fcn [25] proposes a two-dimensional approach to scene text recognition that uses a semantic segmentation network and a word-forming module to recognize text and predict the position of each character. TextScanner [26] has designed two branches, one for character classification and the other for character position and order prediction; by assigning the character positioning to different channels, the characters are naturally separated and sorted. It also proposes a new mutual monitoring mechanism, enabling the two branches to monitor and enhance each other using sequence-level annotations.

CTC-based: CTC-based methods [27,28] use connectionist temporal classification (CTC) to complete sequence recognition, and CRNN [29] uses CNN and RNN as model encoders to extract the text features in images. The CTC decoder is then used to predict the character class for each time step; however, the computation of RNN depends on the length of the input sequence, which takes a long time and can cause gradient disappearance or explosion. Gao et al. [30] replaced RNN with CNN and used a stacked convolution layer to effectively capture the input sequence’s context correlation, which made the computation complexity of the model lower and the parallel computation easier.

Attention-based: SAR [31] is a proposed model for better recognition of irregular text based on a 2D attention module to locate and identify characters individually. DAN [32] solves the problem that traditional attention mechanisms cannot align by decoupling the alignment operation from the history-decoding results. Due to the excellent performance of Transformer in natural language processing, more and more researchers are applying Transformer to STR. ViT [33] shows that Transformer performs significantly better in image classification than deep networks such as Resnet when pre-trained on large datasets and migrated to small and medium image-recognition datasets. It can significantly reduce the computational resources required for training. Deit [34] proves that ViT can achieve a better performance without requiring large datasets.

We not only aim to suppress attention drift but also focus on the speed and calculation cost of the model. Therefore, we designed a portable, fast, and effective model, LSTR, and proposed a position-enhancement module and a visual-enhancement module based on ViT so that the decoding process can automatically adjust the position between the feature area and the target. The experimental results show that our model has excellent competitiveness.

## 3. LSTR Model

To achieve higher accuracy with fewer parameters, we propose a Vision Transformer-based model called LSTR, as shown in Figure 1. In the encoder part, to control the model size, we use Vision Transformer as the backbone network to extract visual information. We also design a position branch to learn the character position information, which helps optimize the spatial alignment between the feature map and the target character area, improving the recognition accuracy and robustness. In the decoder part, we introduce the position-enhancement module (PEM) and visual-enhancement module (VAE), using the position-enhancement module to reinforce the position information and combining the enhanced position information with visual information through the visual-enhancement module. This enhances the model’s perception of text targets, alleviates attention drift, and thus improves the model’s accuracy.

### 3.1. Encoder

The encoder of the model mainly consists of two parts: the backbone network and the position branch. Next, we will provide a detailed introduction.

#### 3.1.1. Backbone

We used a 12-layer ViT and positional branch as the encoder of LSTR, a Transformer structure similar to ViTSTR and DieT, except that we studied the input image size and finally selected the input image size of 32 ∗ 128.

As shown in Figure 2, the input picture $x\in R^{H\times W\times C}$ is divided into a series of non-overlapping blocks $x_{p}\in R^{N\times \left(P^{2}C\right)}$, where *H* and *W* represent the height and width of the image, respectively, *C* the number of channels, P∗P the patch dimension, and *N* the length of the patch sequence.

An encoder consists of *L* encoder blocks. In our model, L=12, and Figure 2 shows one of the encoder blocks. The input of the encoder is: (1)$$z_{0}=\left[x_{Class};x_{P}^{1}E;x_{P}^{2}E;\cdots X_{P}^{N}E\right]+E_{pos}$$
where $x_{Class}\in R^{1\times D}$ is the [class] embedding, $E\in R^{\left(P^{2}C\right)\times D}$ stands for the linear projection matrix, and $E_{pos}\in R^{(N+1)\times D}$ is the positional embedding. The input of each encoder block passes through a layer normalization (LN) followed by a multi-head self-attention layer (MSA), then a multi-layer perceptron (MLP), and finally connects with the residuals. Among them, MSA determines the relationship between eigenvectors, and MLP consists of two linear layers with a Gelu activation function. The equation for the MSA blocks is: (2)$$z_{l}^{\prime }=MSA\left(LN\left(z_{l-1}\right)\right)+Z_{l-1}$$
The term *l* represents the number of encoder blocks, and the MLP equation is: (3)$$Z_{l}=MLP\left(LN\left(z_{l}^{\prime }\right)\right)+z_{l}^{\prime }$$
The final output is embedded in $Z_{l}\in R^{(N+1)\times D}$ for subsequent recognition and as input to the visual-enhancement module.

#### 3.1.2. Position Branch

In the task of scene text recognition, it is crucial not only to extract the visual features effectively, but also to incorporate the text position information. Therefore, we propose a position branch that encodes the text position information using positional encoding.

The model’s position branch encodes the characters’ position in the text using a single heat vector. “1” stands for appearing in the corresponding dimension; otherwise, “0”. Finally, the position embedding is obtained by using the following equations:(4)$$\begin{array}{ccc} PE(pos,2i) & = & \sin\left(pos/10,000^{2i/d_{model}}\right) \end{array}$$(5)$$\begin{array}{ccc} PE(pos,2i+1) & = & \cos\left(pos/10,000^{2i/d_{model}}\right) \end{array}$$
where pos stands for the position information, *i* stands for the character position, and *d* is the dimension. In the PE matrix, the sin variable is added in the even position, and the cos variable is added in the odd position.

This position branch also includes a linear transformation layer and a normalization layer. By transforming and normalizing the positional encoding, the model can effectively utilize the positional information and improve the accuracy of text recognition. In addition, the position-branch design adopts the idea of residual connection, which enables the model to better utilize the previous information and further improve the performance of the position branch.

### 3.2. Decoder

In the decoding process, the position information will become weaker with an increase in the time step, leading to attention drift and lower accuracy. Therefore, as shown in Figure 3, we entirely use the position information and design a position-enhancement module, which uses the enhanced position information as a query vector to input the visual-enhancement module. Finally, the linear layer outputs the prediction characters.

#### 3.2.1. Position-Enhancement Module

The input of the position-enhancement module is the position-embedding output of the position module in the encoder, and the enhanced position information is obtained through multi-head self-attention (MSA), multi-layer perceptron (MLP), residual connection, and layer normalization (LN). For example, in Equations (6)–(8), the query (Q), key (K), and value (V) of MSA are all positional embedding. As shown in Figure 4, the query vector of MSA adopts the upper triangular mask to prevent information disclosure.
(6)$$\mathrm{MultiHead}\left(Q,K,V\right)=Concat\left({\mathrm{head}}_{1},\ldots ,{\mathrm{head}}_{n}\right)W^{0}$$
(7)$${\mathrm{head}}_{i}=\mathrm{Attention}\left(Qw_{i}^{Q},Kw_{i}^{K},Vw_{i}^{V}\right)$$
(8)Q=K=V=PE
where Q,K, and *V* are the query, key, and value, respectively, and $W_{i}^{Q},W_{i}^{K},$ and $W_{i}^{V}$ are the weight matrices for the *i*-th attention head. Concat is a concatenation function, and $W_{0}$ is a trainable parameter matrix.

The output of the multi-head self-attention mechanism is added to the input query to obtain the residual connection result, which is then passed through layer normalization.
(9)x=Q+Dropout(MultiHead(Q,K,V))
(10)$$P^{\prime }=\mathrm{LayerNorm}\left(\frac{\gamma }{\sigma }\left(x-\mu \right)+\beta \right)$$
where μ and σ represent the mean and standard deviation, respectively, of the vector in the sample dimension, and γ and β represent the learnable scaling and shifting factors, respectively. These parameters are learned during the training process.

Next, a feedforward network is used for feature extraction: (11)$$\mathrm{FFN}\left(P^{\prime }\right)=W_{2}\mathrm{ReLU}\left(W_{1}\left(P^{\prime }\right)+b_{1}\right)+b_{2}$$
where $W_{1}$, $W_{2}$, $b_{1}$, and $b_{2}$ are the weight matrices and biases for the feedforward network (FFN).

Finally, the output of the feedforward network is added to the previous result through residual connection and layer normalization, resulting in the enhanced position feature *P*.
(12)$$P=\mathrm{LayerNorm}(P^{\prime }+\mathrm{Dropout}\left(\mathrm{FFN}\left(P^{\prime }\right)\right))$$

#### 3.2.2. Visual-Enhancement Module

The input of the visual-enhancement module in the position-enhanced visual transformer (PSE) consists of the visual information extracted by the Vision Transformer (ViT) in the encoder and the position information enhanced by the position-enhancement module. This input is processed with multi-head attention (MHA), multi-layer perceptron (MLP), residual connections, and layer normalization (LN) to obtain the visually enhanced features with improved positional information. In MHA, the query Q is the position feature P that has been enhanced by the position-enhancement module, and K and V are the visual features extracted using ViT. The interaction between the positional and visual information can effectively alleviate attention drift. Finally, the enhanced features V are obtained according to the enhancement Formulas (6)–(12) in PSE.

Subsequently, the enhanced visual information V is fed into a linear layer to obtain the final prediction probability y. Finally, the cross-entropy loss function is used to calculate the loss.
(13)$$L_{CE}\left(y,t\right)=-\sum_{i=1}^{C}t_{i}\log y_{i}$$
where L is the loss, *y* represents the output of the model, *t* represents the ground truth label, and *C* represents the number of classes of the labels. $t_{i}$ represents the true label value of class *i*, while $y_{i}$ denotes the predicted probability value of class *i*.

## 4. Experimental Analysis

### 4.1. Datasets

Our work involves three datasets: synthetic datasets and real datasets for model training and standard datasets for model validation. We used not only MJ [35] and ST [36], two synthetic datasets (s), but also nine real datasets (R) from the integration from Bautista et al. [37] to extend the data during the training.

1Synthetic datasets: Because of the high cost of labeling datasets, STR did not obtain sufficient datasets early on, so the synthetic datasets ST and MJ were widely used to train models.MJSynth (MJ): It is a dataset of synthetic text images generated by rendering text onto natural images using a variety of fonts, sizes, and colors. The dataset contains over 9 million images, each of which includes one or more instances of synthetic text. The text in the images is diverse, including different languages, scripts, and text styles. The dataset is primarily designed for training and evaluating OCR systems;SynthText (ST): It is another synthetic text dataset created by cropping the text from natural images and pasting it onto new backgrounds. The dataset was originally designed for scene text detection but has since been adapted for OCR. SynthText contains over 7 million synthetic images, each of which contains a single instance of text. The text in the images is mainly in English but includes examples of other languages. Unlike MJSynth, the text in SynthText is embedded in realistic scenes, such as street signs, shop windows, and posters.2Real datasets: With the development of STR, more and more real datasets have been accumulated, and Bautista and others have integrated these datasets, namely ArT [38], Coco [39], LSVT [40], MLT19 [41], RCTW17 [42], OpenVINO [43], TextOCR [44], ReCTS [45] and Uber [46].ArT: A publicly available scene text recognition dataset that contains scene text images extracted from different works of art. The dataset consists of 9300 text images from various works of art and can be used for tasks such as character recognition, text detection, and text recognition;Coco: A widely used computer vision dataset that contains images and annotations for various tasks such as object detection, image segmentation, scene understanding, etc. The scene text recognition task in the COCO dataset refers to identifying text in the image and converting it to a computer-readable text format.LSVT: A large-scale scene text dataset from China that includes 100,000 images divided into training, validation, and test sets. The scene text in the dataset includes regular text, digits, letters, and Chinese characters.MLT19: A large-scale scene text dataset from multiple Asian countries that includes 10,000 images divided into training, validation, and test sets. The scene text in the dataset includes regular text, digits, letters, and Chinese characters.RCTW17: A scene text recognition dataset from China that includes 8940 images from street and outdoor scenes. The scene text in the dataset includes Chinese, English, and digits.OpenVINO: A scene text recognition dataset released by Intel that includes 23,000 images divided into training, validation, and test sets. The scene text in the dataset includes digits, letters, and Chinese characters.TextOCR: A scene text recognition dataset from India that includes 20,000 images from various scenes. The scene text in the dataset includes Indian languages, English, and digits.ReCTS: A scene text recognition dataset from the Hong Kong University of Science and Technology that includes 6000 images from street and outdoor scenes. The scene text in the dataset includes Chinese and English.Uber: A large-scale scene text dataset from Uber that includes 1 million images from different countries. The scene text in the dataset includes digits, letters, and Chinese characters.

As shown in Figure 5, the synthetic images are mostly operated on text, and the background is relatively simple. Compared with the synthetic image, the background of the real image is more complex, with a stronger contrast between light and dark, a larger difference in the background environment, and the partial occlusion and perspective effect of the image. These complex real images can make the model have better generalization and better deal with the text recognition of complex scenes. Therefore, we further verify the performance of the model in real scenes by combining with real datasets.

We evaluated our model using six widely used datasets, categorized into regular and irregular datasets according to their textual characteristics.

Regular datasets:IIIT5K [47]: This dataset contains 5000 text images extracted from Google image search. These images are high-resolution and undistorted, with little noise or background interference;SVT [48]: This dataset contains 804 outdoor street images collected from Google Street View. These images contain text in various fonts, sizes, and orientations, but with little interference and noise. This dataset is typically used to test OCR algorithm performance in recognizing text with diverse fonts, sizes, and orientations;CUTE80 [49]: It contains 80 curved text images and is primarily used to test the performance of OCR algorithms in curved text recognition tasks. Each image contains 3–5 words with varying fonts, sizes, colors, and some noise and interference. This dataset is commonly used to evaluate the performance of curved text recognition algorithms, particularly for the challenging curved text recognition tasks.Irregular datasets:ICDAR2015 (IC15) [50]: This dataset contains many blurred, rotated, low-resolution images, including 2077 and 1811 versions. We chose to use the 1811 version, which discards some extremely distorted images;SVTP [51]: This dataset contains 645 perspective text images extracted from Google Street View. These images contain text in various fonts, sizes, and orientations, but due to perspective distortion, the text lines may appear curved or distorted. This dataset is typically used to test OCR algorithm performance in processing perspective text images;ICDAR2013 (IC13) [47]: This dataset contains 288 curved text images where the text lines are arranged along curved paths. These images contain text in various fonts, sizes, and orientations, as well as some noise and interference. This dataset is typically used to test OCR algorithm performance in processing text images arranged along curved paths.

### 4.2. Experimental Setup

By comparing the effect of text image size on the model, the height and width of the input image are 32 and 128, respectively. We used Deit-tiny as our encoder’s visual feature extraction module and kept the experimental configuration. Our experiment was performed on an Nvidia Tesla V100 GPU system with two graphics cards, a batch size of 240, and an iteration number of 700,000 steps. With the Adadelta Optimizer, the initial learning rate is 1. The learning rate decay strategy is cosine annealing LR.

### 4.3. Evaluation Metrics

In scene text recognition tasks, recognition accuracy is one of the important performance indicators to evaluate the algorithm. For English scene text recognition tasks, the word accuracy (WA) is typically adopted as a word-level accuracy metric to assess the performance of the model. WA refers to the ratio of the number of correctly recognized words to the total number of words. The accuracy demonstrated in subsequent experiments refers to the word accuracy.
(14)$$Accuracy=\frac{Number_{correct}}{Number_{total}}\times 100\%$$

The number of model parameters, computation cost, and forward inference time are important indicators for evaluating model performance. The computation cost of a model is usually represented by the total number of floating-point operations per second (FLOPs), which reflects the computational complexity and speed of the model. In practical applications, the computation cost of the model needs to be controlled within a certain range to ensure the computing speed and power consumption of the application. Additionally, the forward inference time also reflects the speed and efficiency of the model in practical applications. For resource-constrained devices such as mobile devices, the model size and forward inference time need to be controlled within a certain range to ensure the response speed and resource usage of the application. Therefore, lightweight models have become a hot research topic in recent years.

When evaluating the computation cost of a model, the total number of floating-point operations in the model is usually used as an indicator. When evaluating the forward inference time of a model, the average time required for the model to infer a single image is often used as an indicator. This time can be calculated by averaging multiple forward inferences or measured using performance-testing tools. It should be noted that the model inference time may be affected by hardware devices and input image sizes, so these factors need to be considered when evaluating the model performance.

### 4.4. Comparison with Existing Technologies

Our model is built upon DeiT-tiny, which is an improved model that combines knowledge distillation and ViT. The baseline model only uses DeiT-tiny as the backbone network, without adding the position branch, position-enhancement module and visual-enhancement module. As shown in Table 1, we compared our model, LSTR, with the state-of-the-art approach to scene text recognition and summarized the results of each model across six datasets. It can be seen that LSTR with real datasets performs best in each dataset, improving by nearly 3% compared with SOTA. Our model is only 0.8% lower than the SOTA average without real dataset training. However, as Table 2 shows, their model parameters are nearly ten times as fast as ours, and their operating speeds are nowhere near as fast as ours. These data prove that our model is small, accurate, and has a speed advantage. At the same time, compared to the baseline model, our model achieves a 3% improvement in recognition accuracy while maintaining the same level of parameter size and inference speed.

### 4.5. Comparison with Lightweight Models

To determine whether a scene text recognition model is lightweight, factors such as the number of parameters, computational complexity, and inference speed should be considered. Generally, the computational complexity should be controlled to below several billion, while the inference speed should be fast enough to run on resource-limited devices.

In this paper, the model LSTR is compared with several lightweight models. CRNN is a classic model in scene text recognition, while ViTSTR is a simple single-level model architecture that only uses the encoder part of the Transformer and achieves an efficient recognition performance. MGP is an encoder–decoder structure, with the encoder also using Transformer and adopting a multi-granularity prediction strategy to improve the decoder of the model and enhance the recognition accuracy. Due to the size of the encoder, these two models are divided into tiny, small, and base versions, providing lightweight versions (tiny, small) while pursuing high accuracy.

As shown in Table 3, we compare LSTR with these models in terms of performance. Compared with CRNN, LSTR is slightly inferior in model speed and computational complexity, but has a 17.95% higher accuracy than CRNN. Compared with VitSTR, LSTR has higher accuracy than any of its versions, but is not as fast as ViTSTR and has a higher computational complexity. This is because we added the decoder and made corresponding improvements to the decoding part, sacrificing some model speed while increasing some computational complexity, resulting in a 12.25%, 8.15%, and 6.8% increase in recognition accuracy, respectively, significantly improving the performance of our model.

Compared with the lightweight versions of MGP (tiny, small), LSTR has an average accuracy that is 3.45% and 0.8% higher, respectively. Additionally, like MGP, LSTR improved the decoder part. However, our model is not only superior in lightweight modeling but also superior in model accuracy. These data demonstrate the excellence of our model in the realm of lightweight models.

Therefore, under the premise of being the same tiny version, our model achieved the best performance while maintaining the same level of parameters, time, and FLOPS. Compared with other models in the small and base versions, we still achieve a good performance, while also having significant advantages in terms of the parameters, time, and FLOPS.

### 4.6. Ablation Study

In VitSTR and other studies, the image size is 224 × 224, and the patch size is 16 × 16. However, as shown in Table 4, our experiment found that compared with the size of the 224 × 224 image, the size of the 32 × 128 image can retain more essential features of the text image and has more gain on the model. Therefore, our subsequent experiments are based on this.

The results of the ablation experiments in Table 5 show that we improved the accuracy by an average of 0.3% by adding a visual-enhancement module to the baseline, which indicates that the position information is effective for the augmentation of visual information, but that alone is not a significant increase. Therefore, we added a position-enhancement module to give the position branch the learning ability, further optimize the position feature, better represent the space layout between the characters, and ease the problem of attention drift. The experiments show that our recognition rate is improved by nearly 3% compared with the baseline, which shows our improvement is effective.

## 5. Conclusions

In order to improve the attention drift problem of scene text recognition models and control the training costs of the model, we constructed a model called LSTR based on Vision Transformer. First, a position branch is added and an attention mechanism is used to further align the position information and visual information, thereby alleviating the problem of attention drift. Second, it is a simple model structure that uses a tiny version and has only 7 M parameters but achieves a good accuracy, maintaining the lightweight characteristics of tiny models while greatly improving the recognition performance. Compared with the existing baseline models of scene text recognition, LSTR has achieved good results in terms of the accuracy, speed, and cost, making it more applicable and feasible in practical applications, especially in high-demand tasks such as scene text recognition that requires both accuracy and efficiency.

## Figures and Tables

**Figure 1 sensors-23-04490-f001:**
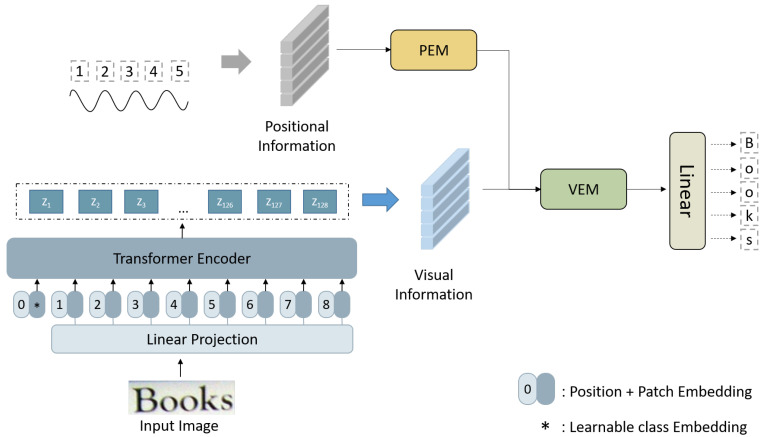
Model architecture of the LSTR.

**Figure 2 sensors-23-04490-f002:**
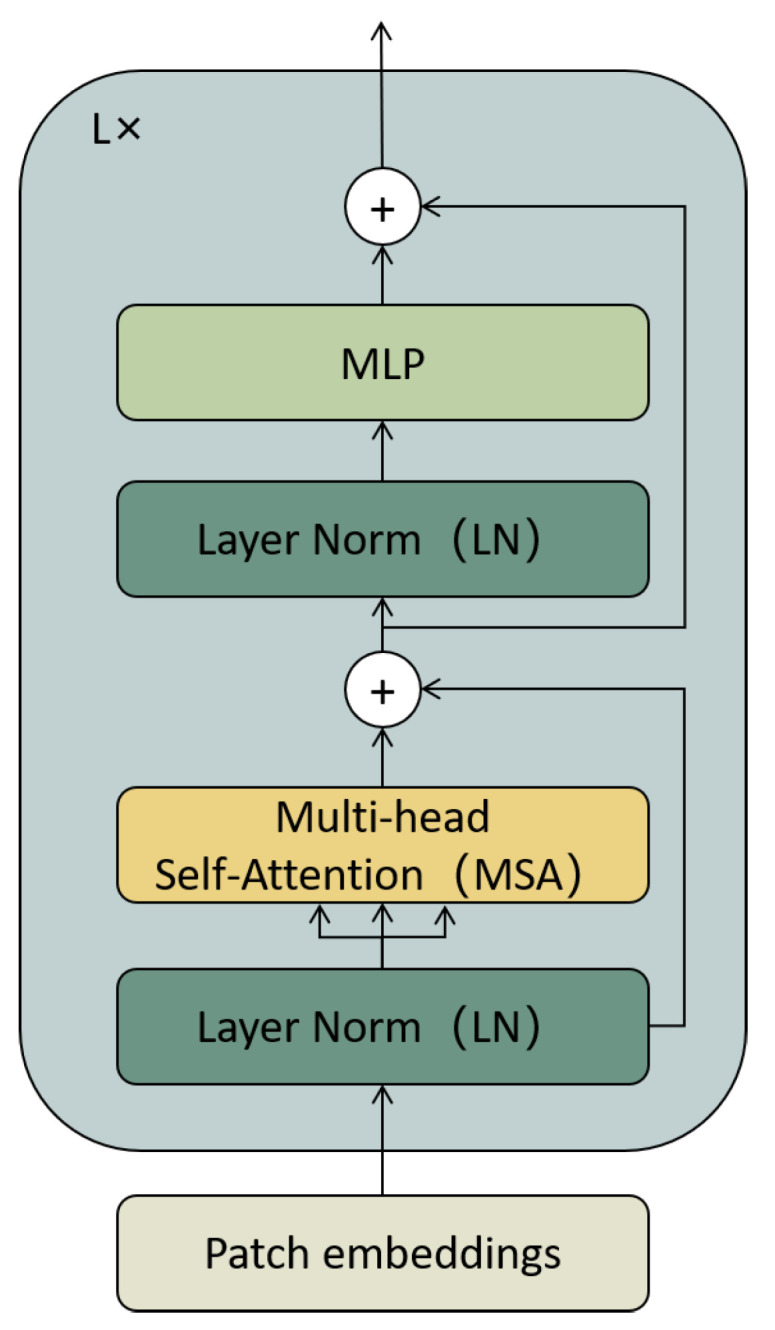
Transformer encoder block.

**Figure 3 sensors-23-04490-f003:**
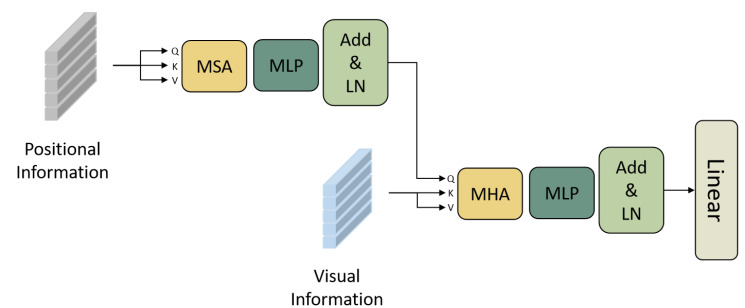
Decoder frame.

**Figure 4 sensors-23-04490-f004:**
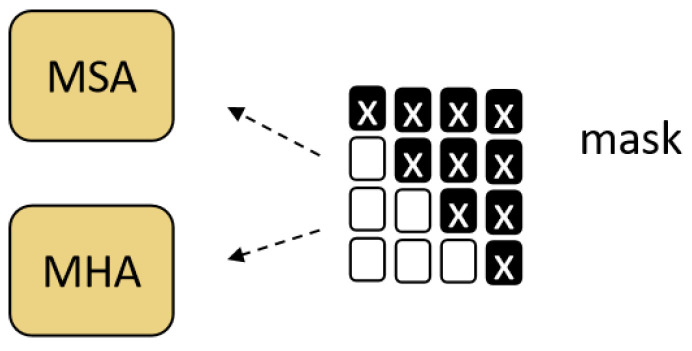
The query vector uses an upper triangular mask, where the ‘X’ values are assigned as ‘-inf’.

**Figure 5 sensors-23-04490-f005:**
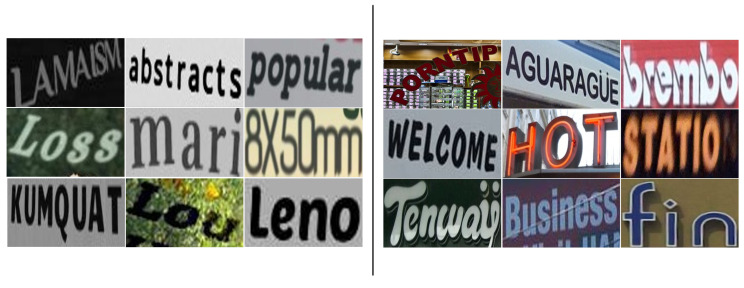
Comparison between synthetic datasets and real datasets. The (**left**) is the synthetic datasets and the (**right**) is the real datasets.

**Table 1 sensors-23-04490-t001:** Comparison of performance with existing technologies.

Method	Year	Train Data	IC13	SVT	IIIT5k	IC15	SVTP	CUTE80	Avg
CRNN	2017	S	89.4	80.1	81.8	65.3	65.9	61.5	74
RARE	2016	S	92.3	85.4	86	73.5	75.4	71	80.6
TRBA	2019	S	93.4	87.6	87.8	77.4	78.1	75.02	83.22
SAR	2019	S	91	84.5	91.5	69.2	76.4	83.5	82.68
VisionLAN	2021	S	95.7	91.7	95.8	83.7	86	88.5	90.23
ViTSTR	2021	S	93.2	87.7	88.4	78.5	81.8	81.3	85.15
ABI	2021	S	97.4	93.5	96.2	86	89.3	89.2	91.93
cdist	2021	S	94.6	93.8	96.5	86.2	89.7	89.58	91.73
SVTR	2022	S	97.2	91.7	96.3	86.6	88.4	95.1	92.55
CornerTransformer	2022	S	96.4	94.6	95.9	86.3	91.5	92	92.78
baseline	-	S	94.86	91.8	93.76	83.26	83.41	86.8	88.98
LSTR	-	S	96.96	94.12	95.6	87.41	88.83	88.83	91.95
LSTR	-	S, R	98.6	96.44	98.13	90.94	92.55	97.22	95.64

**Table 2 sensors-23-04490-t002:** Comparison of the parameters with existing technologies.

Method	Year	Avg	Parameters ($1\times 10^{6}$)	Time (ms/Image)
CRNN	2017	74	8.3	6.3
RARE	2016	80.6	10.8	18.8
TRBA	2019	83.22	49.6	22.8
SAR	2019	82.68	57.5	120.0
VisionLAN	2021	90.23	32.8	28.0
ViTSTR	2021	85.15	85.5	9.8
ABI	2021	91.93	36.7	50.6
cdist	2021	91.73	65.4	123.28
SVTR	2022	92.55	40.8	18.0
CornerTransformer	2022	92.78	85.7	294.9
baseline	-	88.98	5.4	13.6
LSTR	-	91.95	7.1	13.6
LSTR ^*^	-	95.64	7.1	13.6

* Training dataset is S and R.

**Table 3 sensors-23-04490-t003:** Performance comparison with lightweight models.

Method	Year	IC13	SVT	IIIT5k	IC15	SVTP	CUTE80	Avg	Parameters	Time	FLOPS
CRNN	2017	89.4	80.1	81.8	65.3	65.9	61.5	74	8.3	6.3	1.4
ViTSTR-tiny	2021	90.8	93.2	83..7	72.0	74.5	65.0	79.1	5.4	9.3	1.3
MGP-tiny	2022	94.0	91.1	94.3	83.3	83.5	84.3	88.5	21.0	12.0	7.2
LSTR	—	96.96	94.12	95.6	87.41	88.83	88.83	91.95	7.1	13.6	5.7
ViTSTR-small	2021	91.7	87.3	86.6	77.9	81.4	77.9	83.8	21.5	9.5	4.6
MGP-small	2022	96.8	93.5	95.3	86.1	87.3	87.9	91.15	52.6	12.2	25.4
ViTSTR-base	2021	93.2	87.7	88.4	78.5	81.8	81.3	85.15	85.5	9.8	17.6
MGP-base	2022	97.3	94.7	96.4	87.2	91.0	90.3	92.81	148.0	12.3	94.7

The unit of Parameters is ($1\times 10^{6}$), Time is (ms/image), and FLOPS is ($1\times 10^{9}$).

**Table 4 sensors-23-04490-t004:** Comparison of effects of different image sizes on performance.

Method	Image Size (Patch)	IC13_857	SVT	IIIT5k	IC15	SVTP	CUTE80	Avg
baseline	32 × 128 (4 × 4)	94.86	91.80	93.76	83.26	83.41	86.80	88.98
baseline	224 × 224 (16 × 16)	95.79	90.10	94.10	82.71	83.56	83.68	88.32

**Table 5 sensors-23-04490-t005:** Comparison of ablation experiments.

Method	IC13	SVT	IIIT5k	IC15	SVTP	CUTE80	Avg
baseline	94.86	91.80	93.76	83.26	83.41	86.80	88.98
+VisAug	95.21	91.65	94.36	83.43	85.27	85.76	89.28
+posAug+VisAug	96.96	94.12	95.60	87.41	88.83	88.83	91.95

## Data Availability

The data that support the findings of this study are openly available in the public domain.

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
