# Peer review of "Lightweight Scene Text Recognition Based on Transformer"

_sensors, 2023, doi:10.3390/s23094490_

Round 1

Reviewer 1 Report

The paper is reasonably well written and very interesting and useful in the context of  STR. 

My suggestion is to accept the paper for publication.

Only two remarks:

1. pag 5, row 135: I should be i

2. pag 5, equation (6): Mead should be head

Author Response

Dear Reviewer,

Thank you for taking the time to review my paper and providing valuable feedback. Your comments have been extremely helpful in improving my research work.

Regarding your remarks, please find below my response:

Page 5, row 135: "I" should be "I".

Page 5, equation (6): "Mead" should be "head".

I appreciate you bringing these issues to my attention. Based on your suggestions, I have made revisions and provided clearer explanations in the paper.

I will continue to work towards improving my research, and I am grateful for your valuable feedback on my paper.

Thank you once again for your time and efforts in reviewing my paper.

Best regards,

Xin Luan

Reviewer 2 Report

This paper proposes a scene text recognition method based on ViT. However, the main drawback of this paper is the lack of novelty and the details of the network are not clearly described. Meanwhile, there exist issues in this paper that need to be clarified:

1.      Which metric does the “3% higher than the baseline” in the abstract refer to?

2.      The authors should give the full names of PSE and VSE in Figure 1 and describe their structures and effects in detail.

3.      It is suggested to reorganize Section 3. Please describe the modules proposed in this paper in more detail as sub-titles. Meanwhile, the overall network in training and inference processes should be described in more detail.

4.      The authors emphasize that the proposed algorithm is a lightweight model. However, it seems that this paper only chooses a lightweight baseline network without making improvements to make the network efficient. In addition, when measuring the complexity of the network, the calculations of the model should be considered as well as the number of parameters.

5.      Please explain the way to get the position information in Decoder.

6.      Please add the evaluation metrics used in the experiments.

Author Response

We appreciate the feedback you provided for our paper. Our response has been uploaded as an attachment. Please see the attachment.

Reviewer 3 Report

In this paper, the authors proposed a position-enhanced branch in the decoder to mitigate the attention drift and dynamically fuse the position information with the visual information for a better recognition effect. They obtained promissing results. The literature review is very good for this issue. 

I have only one remark. The authors should improve their conclusions.

Author Response

Dear Reviewer,

Thank you for taking the time to review my paper and providing valuable feedback. Your comments have been extremely helpful in improving my research work.

Regarding your remark, please find below my response:

The authors should improve their conclusions.

I appreciate you bringing this issue to my attention. I completely agree with your suggestion, and I have revised the conclusions and provided clearer explanations in the paper.

Overall, your comments have had a profound impact on my paper, making it more comprehensive. I will continue to work towards improving my research, and I am grateful for your valuable feedback on my paper.

Thank you once again for your time and efforts in reviewing my paper.

Best regards,

Xin Luan

Round 2

Reviewer 2 Report

The author has answered and explained all the questions in the revised paper. I have no further questions.